# Epigenetic liquid biopsies reveal endothelial turnover and erythropoiesis in asymptomatic COVID-19

Roni Ben-Ami[1] , Netanel Loyfer[2], Eden Cohen[3], Gavriel Fialkoff[2] , Israa Sharkia[2,4], Sheina Piyanzin[1], Naama Bogot[5], Danit Kochan[6], George Kalak[6], Amir Jarjoui[6] , Chen Chen-Shuali[6], Hava Azulai[6], Hezi Barhoum[6], Nissim Arish[6], Moshe M Greenberger[7] , David Velleman[7], Ramzi Kurd[7] , Eli Ben-Chetrit[7] , Davina Bohm[7], Talya Wolak[7], Ahmad Quteineh[8], Gordon Cann[9], Benjamin Glaser[10], Nir Friedman[2,4,11] , Tommy Kaplan[1,2], Ruth Shemer[1], Ariel Rokach[6], Yuval Dor[1]

**Understanding the full spectrum of tissues affected by SARS-CoV-2 is crucial for deciphering the heterogeneous clinical course of COVID-19. We analyzed DNA methylation and histone modifications in circulating chromatin to assess cell type–specific turnover in patients ranging from asymptomatic to severe cases, in relation to clinical outcomes. Severe COVID-19 was marked by a massive elevation of circulating cell-free DNA (cfDNA) from lung epithelium, cardiomyocytes, vascular endothelium, and erythroblasts, indicating increased cell death or turnover. The immune response was reflected by elevated B-cell and monocyte/macrophage cfDNA and an interferon response before cfDNA release. Strikingly, monocyte/macrophage cfDNA (but not monocyte counts), as well as lung epithelial and endothelial cfDNA, predicted clinical deterioration and duration of hospitalization. Asymptomatic patients had elevated immune cfDNA but no evidence of pulmonary or cardiac damage. Surprisingly, these patients showed elevated endothelial and erythroblast cfDNA, suggesting subclinical vascular and erythrocyte turnover are universal features of COVID-19, independent of disease severity. Epigenetic liquid biopsies provide a noninvasive means of monitoring COVID-19 patients and reveal subclinical vascular damage and red blood cell turnover.**

## Introduction

SARS-CoV-2 infection directly injures the respiratory system and inflicts damage to multiple additional tissues including the heart (Guzik et al, 2020), blood vessels (Pons et al, 2020), pancreas (Goyal et al, 2021), kidneys (Ahmadian et al, 2021), and liver (Amin, 2021). In most of these cases, it is not clear whether the damage is caused directly by the viral infection, or whether tissues suffer damage because of the massive host's immune response. This immune response involves both the innate and adaptive arms of the system (Schulte-Schrepping et al, 2020; Mathew et al, 2020), and is protective in most cases but can often cause severe inflammatory damage (Jose & Manuel, 2020). Despite the extensive information that has accumulated since the emergence of the COVID-19 pandemic, the full spectrum of tissues affected by the infection is still not clear. This may partly explain why we still lack tools to predict the clinical course of disease in infected individuals, which ranges from fatal lung failure to a completely asymptomatic presentation. Clinical heterogeneity during the acute phase can potentially account for variation in late-onset clinical phenotypes such as cardiovascular complications (Xie et al, 2022), new-onset diabetes (Rubino et al, 2020), and the diverse array of long COVID symptoms.

Cell-free chromatin fragments released from dying cells contain extensive information on the identity of cells that released these fragments, and on gene expression programs that operated in the cells before their death. In methylation-based liquid biopsies, the tissue origins of cfDNA are inferred based on its tissue-specific methylation patterns. Because the half-life of cfDNA is extremely short (estimated at 15–120 min) (Lo et al, 1999), such analysis can determine the rate of cell death or turnover in specific host tissues close to the time of sampling. Previous studies characterized the methylome of cfDNA in COVID-19 patients, and deconvoluted it using a partial human cell-type methylome reference atlas

[1]Department of Developmental Biology and Cancer Research, Institute for Medical Research Israel-Canada, Faculty of Medicine, Hebrew University of Jerusalem, Jerusalem, Israel   [2]School of Computer Science, Hebrew University of Jerusalem, Jerusalem, Israel   [3]Department of Military Medicine and "Tzameret", Faculty of Medicine, Hebrew University of Jerusalem, Jerusalem, Israel   [4]Alexander Silberman Institute of Life Science, Hebrew University of Jerusalem, Jerusalem, Israel   [5]Department of Radiology, Hebrew University of Jerusalem, Jerusalem, Israel   [6]Pulmonary Institute, Hebrew University of Jerusalem, Jerusalem, Israel   [7]Department of Medicine, Hebrew University of Jerusalem, Jerusalem, Israel   [8]The Institute of Pediatric Gastroenterology and Nutrition, Shaare Zedek Medical Center, Faculty of Medicine, Hebrew University of Jerusalem, Jerusalem, Israel   [9]GRAIL LLC, Menlo Park, CA, USA   [10]Endocrinology and Metabolism, Hadassah Medical Center and Faculty of Medicine, Hebrew University of Jerusalem, Jerusalem, Israel   [11]Lautenberg Centre for Immunology and Cancer Research, Hebrew University of Jerusalem, Jerusalem, Israel

Correspondence: yuvald@ekmd.huji.ac.il; rokach.ariel@gmail.com

(Andargie et al, 2021; Cheng et al, 2021; Li et al, 2024). The key finding of these studies was that erythroblast turnover is elevated in severe COVID-19 patients and predicts mortality. However, although methylome deconvolution studies can provide an unbiased overview of cfDNA composition, they depend on the quality and breadth of the reference atlas and are typically limited in sensitivity such that tissue contributions to cfDNA amounting to <1% of the total are not detected. To overcome this limitation, we employed deep whole-genome bisulfite sequencing (WGBS) of plasma cfDNA followed by deconvolution using a novel extensive human methylome atlas (Loyfer et al, 2023). In addition, we used a highly sensitive targeted panel of tissue-specific methylation markers to assess tissue turnover in a cohort of COVID-19 patients ranging from very severe hospitalized cases to asymptomatic individuals. Finally, we employed a novel method for cell-free chromatin followed by immunoprecipitation (cfChIP-seq) (Sadeh et al, 2021) to assess gene expression programs in cells of COVID-19 patients before cfDNA release.

## Results

### Patients with severe COVID-19 have a higher concentration of cfDNA, originating in affected tissues

Hospitalized COVID-19 patients (n = 120, total n =142 plasma samples) had a dramatic, 10-fold elevation in the concentration of total cfDNA (Fig 1A), suggesting massive cell death or turnover, or alternatively slower clearance. To identify the tissues responsible for the release of cfDNA, we employed initially an unbiased approach—deep WGBS (57x coverage in average)—on six plasma samples obtained from hospitalized, severe patients infected during summer 2020 and six samples from age- and sex-matched healthy controls. All samples came from unvaccinated individuals.

Deep WGBS was followed by unsupervised deconvolution based on an extensive reference atlas of human tissue-specific methylomes of 37 cell types (Loyfer et al, 2023), allowing highly accurate identification of methylation patterns derived from all these potential sources (Loyfer et al, 2023). The composition of cfDNA in the healthy controls is consistent with previous studies. Blood cell types account for 91% of the cfDNA (granulocytes 44%, megakaryocytes 24%, monocytes/macrophages 18%, NK cells 2%, and B cells, T cells, and erythroblasts each contributing ~1%), and the rest originated from vascular endothelial cells (7%) and hepatocytes (2%) (Fig 1B and C). Consistent with previous reports (Moss et al, 2018; Magenheim et al, 2022), we do not find evidence of DNA derived from the lung in healthy individuals. In contrast, COVID-19 patients had an elevation in the fraction of cfDNA that originated from the liver (3%), T cells (4%), B cells (8%), erythroblasts (12%), and lungs (2%), at the expense of the relative contribution of granulocytes (mainly neutrophils), megakaryocytes, and monocytes/macrophages (32%, 18%, and 15%, respectively) (Fig 1B and C). When considering both the relative contribution of each tissue and the total concentration of cfDNA, we observed an elevation in the concentration of cfDNA derived from all blood

lineages, in addition to vascular endothelial cells, hepatocytes, and lung epithelial cells (Fig 1D).

These findings provide an unbiased view of cell turnover in severe COVID-19, which is consistent with previous reports and with the clinical nature of the disease. Elevated lung cfDNA likely reflects excessive lung cell death or turnover, which is expected given that the lung is the primary site of infection and the key contributor to clinical damage. Erythroblast cfDNA was previously shown to be elevated in COVID-19 patients, although it was suggested that only morbidly sick patients show this phenomenon, whereas we identify a massive elevation in erythroblast cfDNA in 4 of 6 patients (see below further details using targeted analysis). Similarly, elevated cfDNA from immune cells is to be expected, as a reflection of the normal immune response of the host. Vascular endothelium–derived cfDNA may reflect the documented evidence of vascular damage in patients with severe COVID-19 (Iba et al, 2020).

### A targeted assay for cfDNA biomarkers relevant to COVID-19 patients

To further characterize cfDNA content in COVID-19 patients, we applied a targeted cfDNA methylation assay on the entire cohort of plasma samples. This allows for ultra-deep assessment of specific targets at a modest cost. The use of targeted markers allows to sequence essentially every molecule from a given marker locus, providing higher sensitivity without compromising specificity. We designed a panel of 37 methylation markers, containing loci that are specifically unmethylated or specifically methylated in cell types relevant to COVID-19 patients. The panel contained markers of specific immune cell types: B cells (four independent markers), T cells (2), CD8+ T cells (1), NK cells (2), monocytes/macrophages (3), and neutrophils (3). In addition, it included methylation markers of other relevant blood cell types including erythroblasts (5) and megakaryocytes (2), as well as markers of lung epithelial cells (6), vascular endothelial cells (4), and cardiomyocytes (5). We validated the specificity of these markers (Fig S1) and established a protocol for two multiplex PCR conditions that could amplify all markers from the cfDNA extracted from each plasma sample (see the Materials and Methods section). The WGBS and the targeted assay findings were in good agreement, as shown is Fig S2.

With this targeted assay at hand, we moved to assessment of samples from the larger cohort of 120 hospitalized patients and 40 controls, all of whom were unvaccinated. Consistent with the methylome deconvolution analysis, we observed a highly significant increase in cfDNA originating in immune cells, including elevated levels of cfDNA from B cells, T cells, monocyte/macrophages, neutrophils, CD8 T cells, and NK cells (Fig 2A). In addition, cfDNA of erythroblasts and megakaryocytes was dramatically elevated, reflecting increased turnover and production of red blood cells and platelets (Fig 2B). We also noticed elevated levels of cfDNA derived from lung epithelial cells and vascular endothelial cells, consistent with the deconvolution analysis (Fig 2B). Importantly, some (but not all) patients had significantly elevated levels of cardiomyocyte cfDNA, suggesting ongoing cardiac damage (Fig 2B), consistent with reports on heart damage in severe COVID-19 patients (Guzik et al, 2020; Szekely et al, 2020).

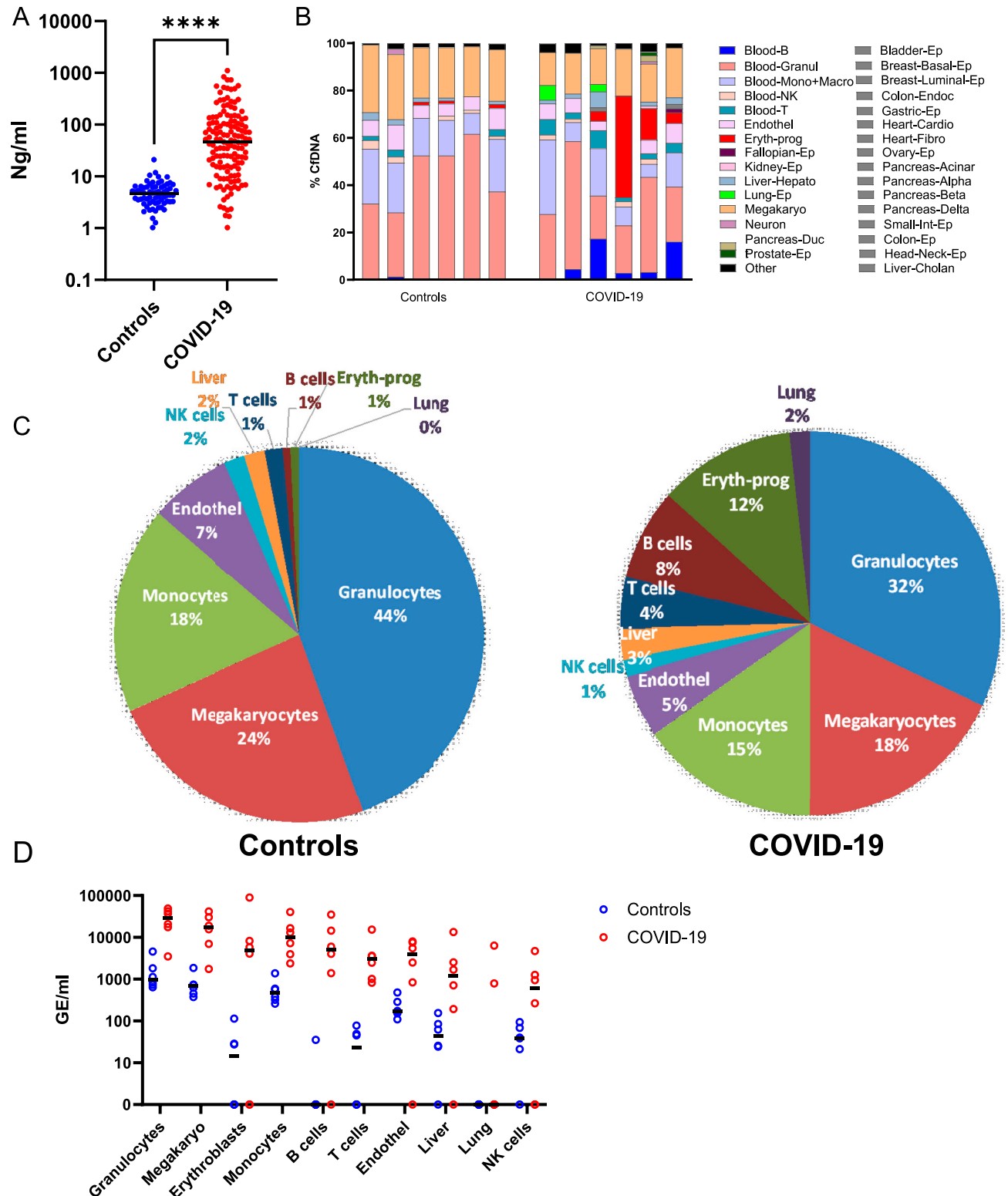

**Figure 1. Whole-genome bisulfite sequencing analysis of severe COVID-19 patients and sex- and age-matched controls.**
**(A)** CfDNA concentration of hospitalized patients (n = 120) and controls (N = 68). **(B)** Deconvolution of plasma samples from severe patients (N = 6) and matched controls (N = 6). Each column is a single sample. These samples are used for further analysis in the next panels. **(C)** Average relative contribution of indicated cell types in patients and controls. **(D)** Absolute values of cfDNA from indicated cell types in patients and controls, derived by multiplying the fraction of cell type–specific cfDNA by the total concentration of cfDNA in the sample. Values are expressed as genome equivalents per ml plasma. Each dot represents one sample. *P < 0.05, **P < 0.01, ***P < 0.001, and ****P < 0.0001. Band indicates the median value.

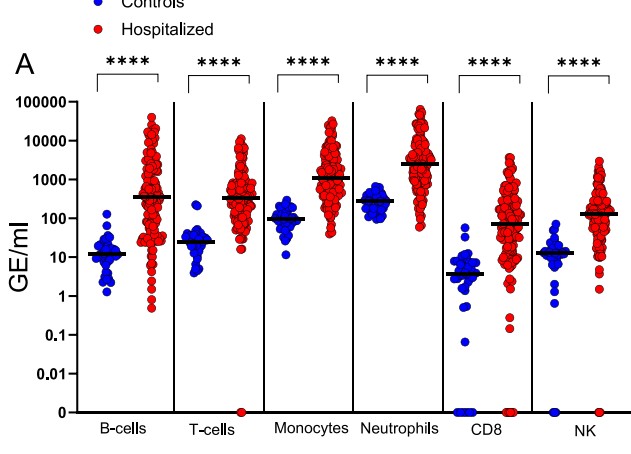

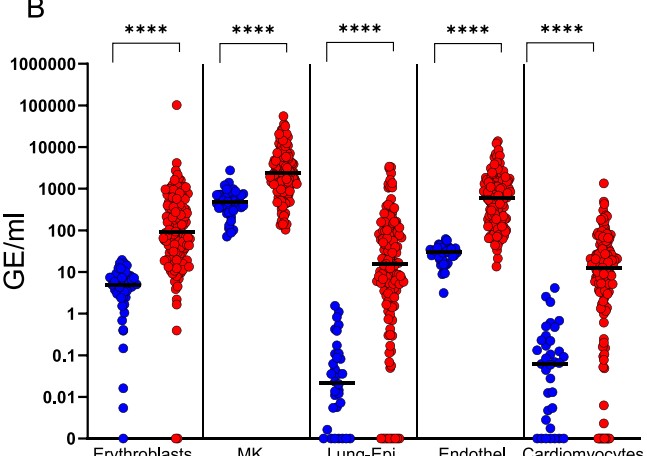

**Figure 2. Targeted analyses of cfDNA origins in hospitalized patients.**
Each dot represents a sample. In total, the analysis includes 120 hospitalized patients and 30–45 healthy controls. **(A)** Immune-derived cfDNA markers. **(B)** cfDNA methylation markers of erythroblasts, megakaryocytes (MK), lung epithelial cells, vascular endothelial cells, and cardiomyocytes. Note that the median level of cardiomyocyte cfDNA is 12 GE/ml, potentially explaining why this was not detected by the less-sensitive deconvolution analysis. *$P < 0.05$, **$P < 0.01$, ***$P < 0.001$, and ****$P < 0.0001$. Horizontal lines indicate the median value. Because of the nonzero limit of log graphs, we consider 0.001 GE/ml as undetected.

Thus, targeted analysis of cfDNA in hospitalized COVID-19 patients reveals an extensive immune response, and elevated turnover of red blood cells, megakaryocytes, lung epithelial cells, and vascular endothelial cells. In addition, the high sensitivity of the targeted assay showed striking evidence of cardiomyocyte death in hospitalized patients.

## cfDNA correlates to clinical severity score

To understand the potential of cell type–specific cfDNA to explain clinical phenomena, we generated a correlation matrix quantifying the relationship between cfDNA parameters and key clinical records for the entire set of 120 patients hospitalized with COVID-19. The matrix included age and sex, standard biochemical markers, blood cell counts, and the WHO progression scale of COVID-19

clinical severity (WHO Working Group on the Clinical Characterisation & Management of COVID-19 infection, 2020), as well as cfDNA parameters.

To verify internal consistency, we first determined the correlation between non-cfDNA parameters. Consistent with previous studies (Henry et al, 2020; Terpos et al, 2020), clinical severity was correlated with neutrophil counts, and negatively correlated with lymphocyte counts (i.e., associated with lymphopenia). Biochemical measures of stress and inflammation (CRP, D-dimer, and ferritin) also correlated with clinical severity of COVID-19, as well as high neutrophil counts and low lymphocyte counts (Fig 3).

The concentration of cfDNA from immune cells, erythroblasts, heart, lung, and vascular endothelial cells was positively and significantly correlated to disease severity, measured by the WHO score at the time of sampling. Vascular endothelial cell cfDNA showed the strongest correlation to disease severity, followed by the total concentration of cfDNA, neutrophil cfDNA, B-cell cfDNA, and lung epithelium cfDNA (Fig 3). Note that the correlation of cfDNA to disease severity was generally much stronger than the correlation of circulating blood cell counts and all biochemical markers to disease severity, highlighting the direct relationship between clinical condition and tissue damage as reflected by cfDNA.

### CfDNA as a prognostic marker for clinical course

To assess the utility of tissue-specific cfDNA in predicting the clinical trajectory of COVID-19, we generated a simple score of the future clinical course. This score considers the maximal WHO score achieved during hospitalization (until discharge or death), minus the WHO score at the day of plasma sampling. With this, each individual sample is assigned a "recovering" (≤0) or "deteriorating" (0<) status based on the clinical course of the patient.

As shown in Fig 4A, hospitalized patients that have clinically deteriorated in the days after blood sampling had at the time of sampling significantly higher levels of cfDNA from multiple immune cell types including neutrophils, monocytes/macrophages, and NK cells, and to a lesser extent T cells. Interestingly, B-cell cfDNA was not a predictor of clinical outcome, despite B cells being a critical determinant of humoral immunity. This finding, along with the weak correlation between T-cell cfDNA and clinical outcome, suggests that deterioration of hospitalized patients is dictated mostly by performance of their innate immune system, combined with the extent of damage to vital organs (see below) (Schulte-Schrepping et al, 2020; Schultze & Aschenbrenner, 2021).

The correlation of monocyte cfDNA levels to clinical deterioration was particularly strong. Strikingly, the counts of circulating monocytes were not associated with clinical course, suggesting that monocyte/macrophage cfDNA levels reflect the amount of monocyte damage or turnover taking place within tissues (potentially a result of infection; see the Discussion section), without altering systemic cell counts (Fig 4B and C).

Among nonimmune cfDNA, lung epithelial cells and vascular endothelial cells were significantly associated with future deterioration of hospitalized patients, whereas cardiomyocyte, megakaryocyte, and erythroblast cfDNA were not (Fig 4D). Although the correlation between clinical course and the extent of lung

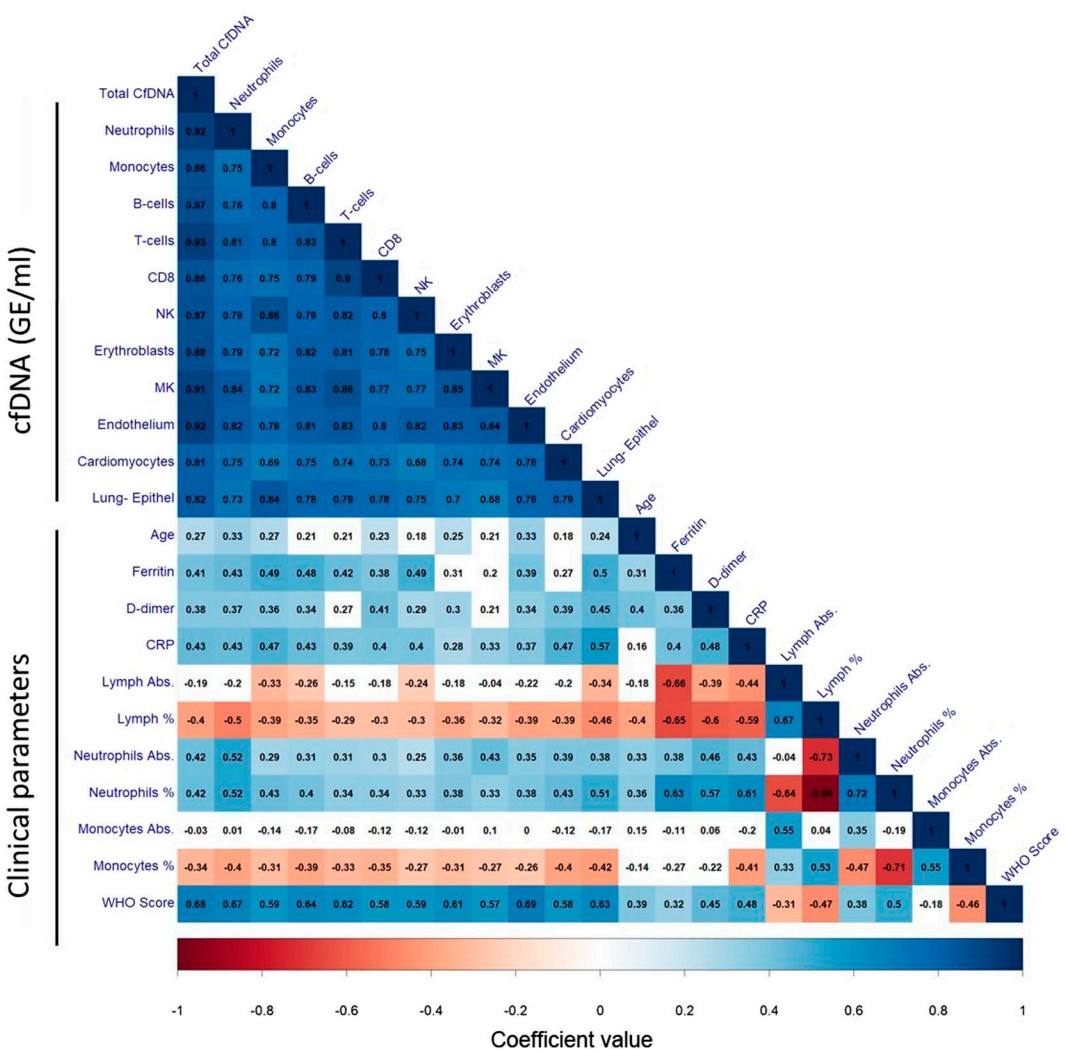

**Figure 3. Correlations between cell type–specific cfDNA, clinical and laboratory parameters, and WHO clinical score.**
Correlation coefficients are presented in black inside each square. Only statistically significant correlations (q < 0.05) are colored, and the blue–white–red color scale reflects the coefficient value.

damage as reflected in lung cfDNA is expected, the relationship between vascular endothelial cell turnover and clinical outcome is less trivial, and indicates a prominent involvement of vascular damage in the pathogenesis and course of COVID-19 (see below).

### Asymptomatic patients show subclinical elevation of vascular endothelial cell turnover and erythropoiesis

We characterized cfDNA in patients with asymptomatic or mild disease, who donated blood while being quarantined because of a positive SARS-CoV-2 PCR test. This is an understudied population, which can presumably provide insights into the cfDNA manifestation of a successful immune response, and into uncovering subclinical damage to organs. The cfDNA concentration in these individuals (n = 19 patients, 8.6 ng/ml) was significantly higher than in healthy controls (n = 68 donors, 4.7 ng/ml) (*P*=0.0001), yet it was far lower than the concentration of cfDNA in the hospitalized

patients described above (n = 120 patients, n = 142 samples, 47.1 ng/ml) (*P*=0.0001) (Fig 5A).

The asymptomatic patients had elevated levels of cfDNA from all immune cell types measured, likely reflecting cellular turnover associated with the successful mounting of an immune response to curb the virus (Fig 5B). The much higher concentration of immune-derived cfDNA in hospitalized patients reflects either increased activity needed to fight off a higher viral load, or alternatively a pathologic overactivity (Dadras et al, 2022).

Asymptomatic patients did not have higher-than-normal levels of megakaryocyte cfDNA, lung cfDNA, or cardiomyocyte cfDNA (<1 GE/ml from lung and cardiomyocytes). However, these patients showed significantly elevated cfDNA levels derived from erythroblasts and from vascular endothelial cells, even though there was no clinical indication of disruption of red blood cell homeostasis or damage to the vasculature (Fig 5C). This finding, in addition to elevation of the d-dimer protein (Fig 3 and references Rostami &

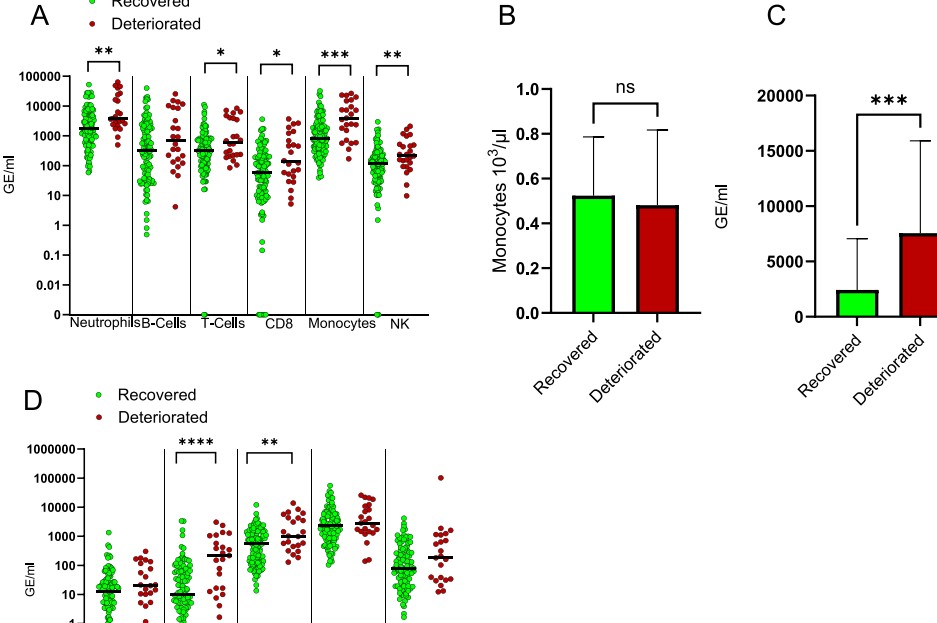

**Figure 4.   Correlation between cell type–specific cfDNA and future clinical course.**
WHO score on the day of sampling was subtracted from the maximal WHO score during the hospitalization, following the sampling day. Nonpositive values were scored as clinical recovery, whereas positive values were scored as deterioration. **(A)** Immune-derived cfDNA levels in patients that recovered or deteriorated clinically after sampling. **(B)** Monocyte cell counts in patients that deteriorated or recovered. **(C)** Monocyte/macrophage-derived cfDNA in patients that deteriorated or recovered. **(D)** cfDNA from cardiomyocytes, lung epithelial cells, vascular endothelial cells, megakaryocytes (MK), and erythroblasts in patients that recovered or deteriorated clinically after sampling. *$P$ < 0.05, **$P$ < 0.01, ***$P$ < 0.001, and ****$P$ < 0.0001. The band indicates the median value. Because of the nonzero limit of log graphs, we consider 0.001 GE/ml as undetected.

Mansouritorghabeh [2020]; Wool & Miller [2021]) and increased nucleated RBC counts in severe COVID-19 (Huerga Encabo et al, 2021), suggests subclinical vascular damage compensated by endothelial cell regeneration, as well as enhanced rate of erythropoiesis (see the Discussion section).

## Circulating chromatin indicates interferon response in dying cells

The course of SARS-CoV-2 infection involves major physiological processes including a strong immune response and cytokine storm as mentioned previously. These processes entail the activation of gene expression programs that are inactive in the normal state of cells. We hypothesized that some of these programs occur in the dying cells and might be reflected in cfDNA. In order to test this, we made use of cfChIP-seq, a recently introduced method of chromatin immunoprecipitation of cell-free nucleosomes carrying active chromatin modifications, which allows to infer gene expression in cells undergoing apoptosis, before the release of cfDNA (Sadeh et al, 2021).

Specifically, we used cfChIP-seq with an antibody against histone 3 lysine 4 trimethylation (H3K4me3), a chromatin modification that marks open promoters (Barski et al, 2007; Heintzman et al, 2007). We applied cfChIP-seq to plasma samples from hospitalized COVID-19 patients (n = 33), asymptomatic COVID-19 patients (n = 13), and a healthy control group (n = 14), with an average yield of ~3.5 million unique reads per sample, which are a genome-wide representation of open promoters, similar to RNA-seq gene expression counts.

Using statistical tests designed for cfChIP-seq, we searched for genes with differential coverage between the hospitalized and healthy samples (see the Materials and Methods section). Several dozen genes came up as significantly elevated in the hospitalized group (Fig S3 and Table S1). Inspection of these genes revealed three clusters with distinct behaviors (Fig 6A). The first cluster contains genes that are expressed only in the liver (e.g., *SERPINC1*, *CFHR5*), and the entire set of genes is strongly enriched for the liver (see enrichment of gene sets for all clusters in Tables S2, S3, and S4). The genes in this cluster were especially high in two of the hospitalized patients but were observed also in other hospitalized and asymptomatic patients, suggesting elevated death of hepatocytes in these cases. The second cluster includes the erythroblast-specific glycophorin genes (*GYPA*, *GYPB*, *GYPE*), and was observed in a different subset of the hospitalized samples, as well as one of the asymptomatic samples. This signal presumably reflects elevated turnover of red blood cells and increased rate of erythropoiesis, and is consistent with the methylation-based observation of elevated erythroblast cfDNA (Figs 1 and 2).

The third cluster, in contrast to the first two, does not reflect the death of a specific cell type but rather a gene expression program that is typically inactive in cells. The genes in this cluster are enriched for the interferon response and include genes such as *IFIT1*, *IFI6*, and *IFI27*.

Computing the cumulative signal over a predefined interferon response gene set from the Human MSigDB Collections (Liberzon et al, 2015) (Table S5), we observed a significant signal in the hospitalized and asymptomatic groups compared with the healthy samples (Wilcoxon's test, $P$ = 8.2 X $10^{-6}$ and $P$ = 0.00042,

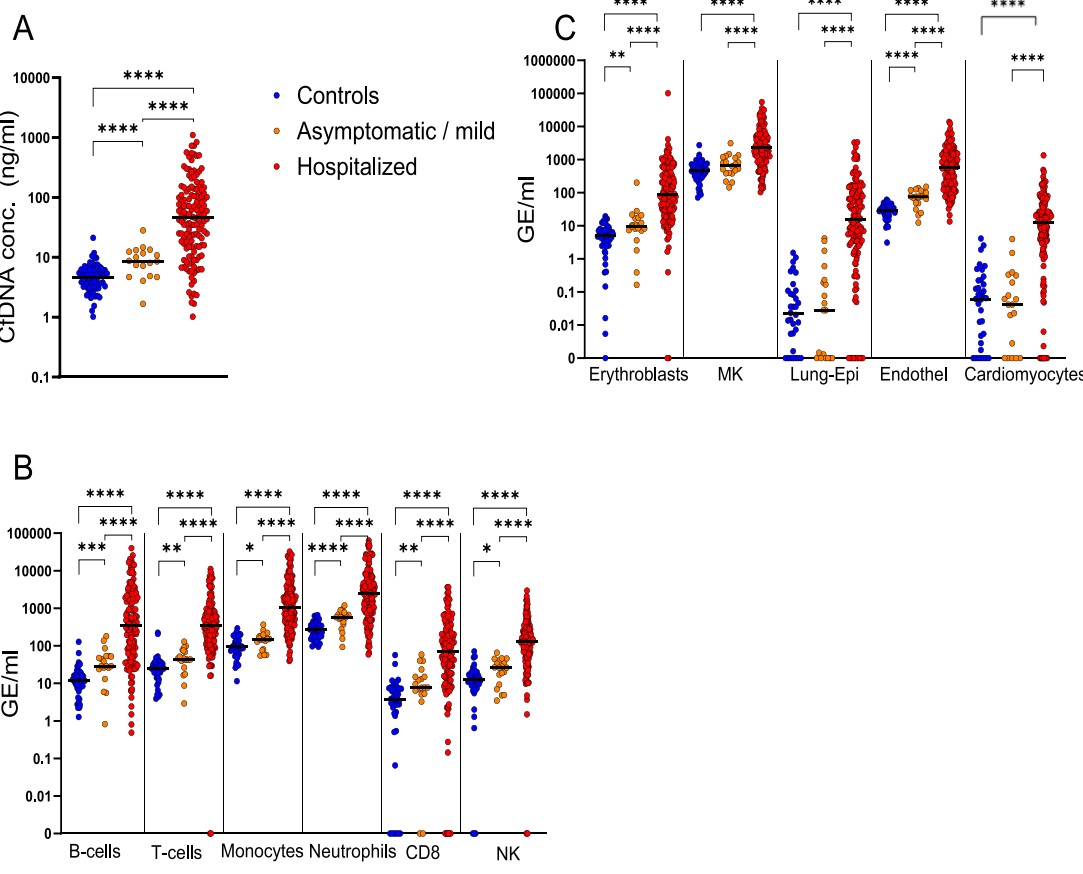

**Figure 5. Cell type–specific cfDNA in patients with asymptomatic or mild COVID-19 (n = 19) compared with hospitalized patients (n = 120) and healthy controls (n = 45).**
**(A)** Total cfDNA concentration. **(B)** Immune-derived cfDNA. **(C)** cfDNA from erythroblasts, megakaryocytes, lung epithelial cells, vascular endothelial cells, and cardiomyocytes. *P < 0.05, **P < 0.01, ***P < 0.001, and ****P < 0.0001. The band indicates the median value. Because of the nonzero limit of log graphs, we consider 0.001 GE/ml as undetected.

respectively), which is particularly pronounced in a subset of samples from hospitalized patients (Fig 6B).

Overall, these results uncover an interferon response occurring in the dying cells of COVID-19 patients.

# Discussion

### Unbiased and targeted epigenetic liquid biopsies in COVID-19

Analysis of cell-free DNA methylation and histone modifications enables the characterization of cellular turnover or death occurring within organs, as well as the gene expression programs active in cells before death and release of cfDNA. In the case of solid organs, this is equivalent to a standard biopsy, with the advantage that information is summed across the entire organ. In the case of the immune cells, cfDNA informs on immune and inflammatory processes taking place within organs, which are not reflected in blood cell counts (Fox-Fisher et al, 2021). Here, we applied this technology in three forms: an unbiased deconvolution of the plasma methylome, targeted analysis of cell type–specific methylation markers, and cfChIP-

seq. An unbiased approach to cfDNA methylation of COVID-19 patients was previously reported by several studies (Andargie et al, 2021; Cheng et al, 2021; Li et al, 2024). Our work differs in several aspects. First, we sequenced plasma at a coverage more than an order of magnitude deeper (57x versus 1.3X [Cheng et al, 2020 *Preprint*]). Second, we used a new, more extensive human cell-type methylome atlas that allows for more accurate interpretation of cfDNA methylation patterns (Loyfer et al, 2023). Third, the targeted assay, while limited to a predefined number of loci, potentially allows for a more accurate and sensitive detection of cfDNA from a given source because of high depth of sequencing (although this idea requires an experimental test). Fourth, we applied the cfChIP-seq assay, which provides for the first time information on gene expression programs in COVID-19 patients, specifically in cells that are turning over and releasing cfDNA.

### Tissue dynamics in COVID-19

The analysis of cfDNA methylation and chromatin patterns led to several insights regarding disease dynamics in COVID-19 patients:

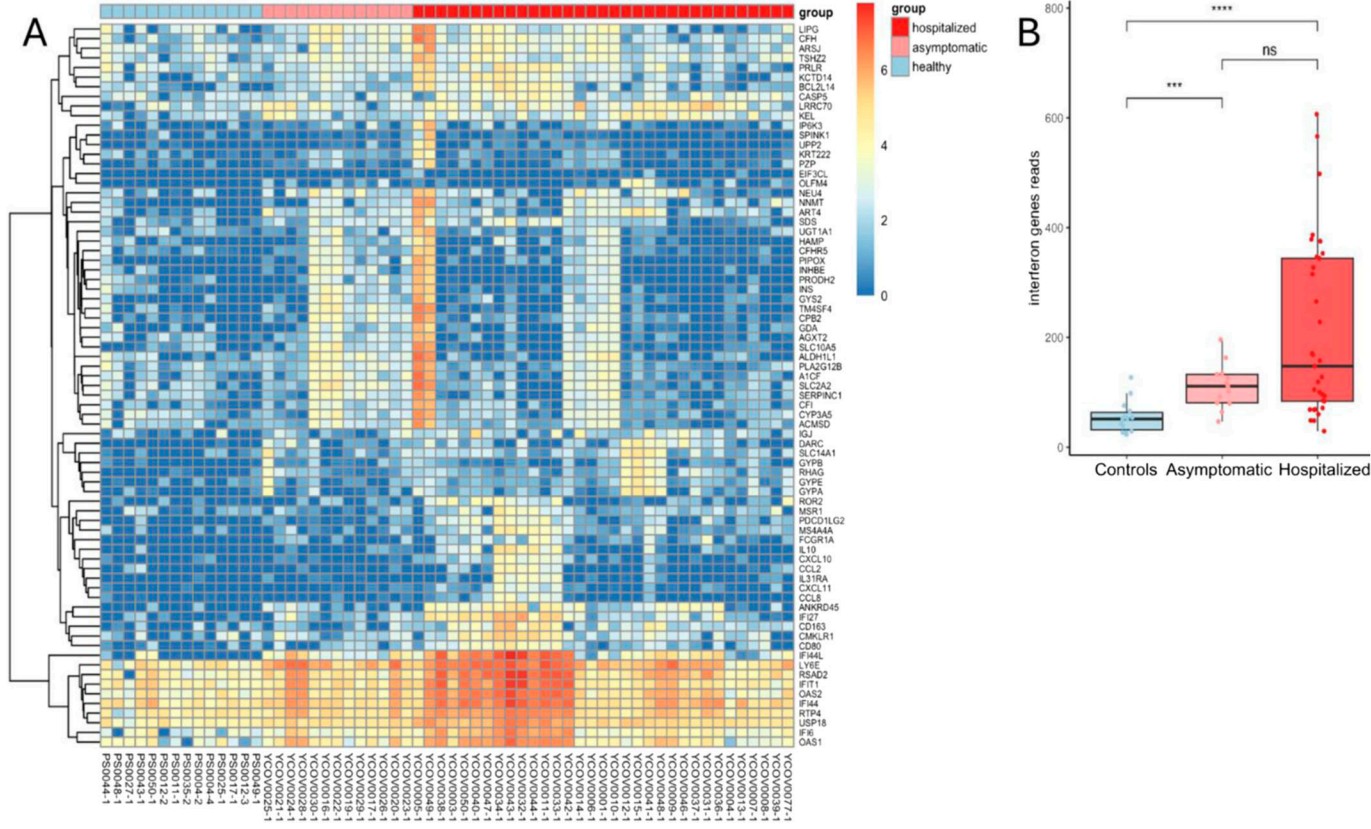

**Figure 6. Chromatin immunoprecipitation of cell-free nucleosomes analysis of COVID-19 patients, asymptomatic patients, and controls.**
**(A)** Unsupervised clustered heat map of the genes that are significantly elevated in the hospitalized COVID-19 patients, compared with controls. Color represents $\log_2$ (1+ normalized promoter coverage). **(B)** Reads of genes from a predefined interferon response gene set in the three groups.

(i) Evidence of frequent cardiomyocyte death in hospitalized patients. Although cardiac damage is a known component of COVID-19 pathology (Chung et al, 2021), it has not been appreciated that cardiac cell death is a feature shared by most hospitalized patients. Our findings are consistent with elevated troponin recorded in COVID-19, and provide evidence that this elevation is the result of true cardiac cell death and not other processes such as cytoplasmic troponin leak or slower renal clearance (Giustino et al, 2020; Sandoval et al, 2020; Fridlich et al, 2023). We note, however, that the magnitude of the phenomenon is small (median of 12 cardiac genome equivalents per milliliter plasma is patients). Further study is required to determine the long-term clinical significance of elevated cardiomyocyte cfDNA in hospitalized COVID-19 patients.

(ii) A strong correlation between disease severity and general cfDNA concentration, as well as the contribution of specific immune cell types and solid organs. Almost all cfDNA parameters measured showed a statistically significant, positive correlation to the standard WHO score of COVID-19 severity. Notably, the correlation of severity to cfDNA parameters was stronger than the correlation to all standard biomarkers including CRP, ferritin, d-dimer, and lymphocyte counts. This underscores the direct relationship between tissue damage and cfDNA. We propose that the short half-life of cfDNA (15–120 min, compared with hours or days for standard biomarkers), a feature that makes it an acute biomarker for tissue turnover at the time of sampling, further strengthens the correlation to the current clinical condition of a patient.

(iii) Prediction of clinical deterioration by the levels of cfDNA derived from innate immune cells, lung epithelium, and vascular endothelial cells. This finding is consistent with extensive literature on the fundamental role of the innate immune system in setting the clinical course of COVID-19. The correlation of monocyte/macrophage cfDNA, but not monocyte counts, to clinical deterioration is particularly intriguing. It is in agreement with a recent report on the prominent role of monocytes in critically ill patients, including the effect of monocyte-expressed genes on vascular permeability in the lung (Pairo-Castineira et al, 2023). One potential interpretation is that elevated monocyte cfDNA reflects direct viral infection of monocytes (Junqueira et al, 2022), which is indicative of viral spread and immune failure. Alternatively, the abnormally high turnover of monocytes may reflect pathogenic activation of the immune system that inflicts tissue damage beyond direct damage because of infection. We acknowledge a

current limitation in the specificity of our markers, which are unable to distinguish between DNA originating in circulating monocytes and DNA originating in tissue-resident macrophages. Further refinement of the methylome atlas may resolve this ambiguity.

(iv) Heightened interferon response. SARS-CoV-2 infection causes a massive elevation in the expression of interferon-stimulated genes, as measured in leukocytes and in material from invasive biopsies (Wilk et al, 2020; Youk et al, 2020; Bhat et al, 2022). The identification of an interferon response in cell-free chromatin underscores the potential of epigenetic liquid biopsies to reveal gene expression program in cells before their death. Further refinement of the cfChIP-seq technology (e.g., analysis of chromatin marks characteristic of tissue-specific enhancers) may allow identification of the cell types that have activated the interferon response before cell death.

(v) Subclinical elevation of erythroblast and vascular endothelial cell cfDNA in patients with asymptomatic or mild disease. The elevated levels of erythroblast cfDNA observed in our experiments are consistent with previous reports that erythroblast cfDNA and counts of nucleated red blood cells are elevated in premorbid COVID-19 patients (Cheng et al, 2021; Düz et al, 2023; Schmidt et al, 2024), and support the idea that SARS-CoV-2 infection causes a disruption of erythropoiesis (Huerga Encabo et al, 2021). Importantly, the higher sensitivity of the targeted assay allowed the detection of mild though significant elevation of erythroblast cfDNA in most of the patients, including severe patients that eventually survived and, most strikingly, asymptomatic patients. This suggests that SARS-CoV-2 infection causes a profound disruption of erythropoiesis, regardless of disease severity or clinical manifestation. A higher viral load and clinical deterioration may increase further the magnitude of this phenomenon. Because red blood cell counts do not change in most COVID-19 patients (Henry et al, 2020), elevated levels of erythroblast cfDNA likely mean a higher rate of turnover. We speculate that SARS-CoV-2 infection shortens the lifespan of red blood cells, triggering a compensatory increase in erythropoiesis which is manifested in elevated erythroblast cfDNA. Additional studies are needed to test this hypothesis and understand the underlying mechanisms. In addition to erythroblast cfDNA, vascular endothelial cell cfDNA levels were significantly elevated in the asymptomatic patients. Although vascular damage was shown to take place in a rhesus macaque model of COVID-19 (Aid et al, 2020), evidence in human patients is more limited because of the paucity of circulating biomarkers of vascular endothelial damage. The finding that even patients with asymptomatic disease have elevated levels of vascular endothelial cfDNA suggests that COVID-19 causes a subclinical increase of vascular turnover, via either direct viral damage or via immune mediators. As long COVID can develop even in patients that had a relatively mild disease, it is particularly interesting to determine whether cfDNA abnormalities in mild cases are prognostic of long COVID.

In summary, epigenetic liquid biopsies open a noninvasive window into tissue dynamics in patients with COVID-19, including subclinical damage and dynamic processes that cannot be assessed via existing biomarkers. Further studies of patients with long COVID will reveal the prognostic and diagnostic utility of cell-free DNA and chromatin in this disease. Beyond COVID-19, our findings underscore the potential of epigenetic liquid biopsies to characterize complex pathologies that do not involve the shedding of mutant cfDNA.

## Limitations of the study

This study has several notable limitations. First, the WGBS atlas consists of 37 cell types that we were able to isolate (Loyfer et al, 2023) and is missing some cell types that were previously mentioned as involved in COVID-19 pathogenesis such as Schwann cells (Loy et al, 2023). Ongoing expansion of the atlas may help to refine the interpretations of WGBS. Second, because of resource limitations, deep WGBS was done on only six critically ill patients, to uncover the potential cell types that will be of most interest; the vast majority of samples were analyzed only using the more affordable targeted assay, which has a deeper coverage and therefore potentially greater sensitivity but addresses a small number of cell types. Third, most of the analyzed samples are of hospitalized COVID-19 patients, with a relatively small number of asymptomatic patients; hence, our conclusions on cfDNA dynamics in asymptomatic patients warrant additional studies. Fourth, to ensure that control patients did not have COVID-19, we used plasma samples collected before February 2020, most of which from young individuals. There was also a sex bias in that the asymptomatic COVID-19 patients were mostly males. We note, however, that in previous studies, cfDNA levels and tissue origins did not differ substantially between sexes and ages (Fox-Fisher et al, 2021). In addition, in the WGBS samples we did match for age and sex. Fifth, different collection methods (e.g., Streck versus EDTA tubes) may lead to potential preanalytical confounders, although in our experience these have relatively minor effects on cfDNA concentration and methylation patterns.

Lastly, our understanding of cfDNA biology remains limited, which impacts interpretation of results. The determinants of cfDNA release and clearance are not fully understood, although cfDNA levels are strongly correlated with tissue pathology. We acknowledge that elevated concentration of cfDNA from a given tissue may represent pathological processes other than cell turnover or death. For example, elevated lung cfDNA in patients may result from vascular damage leading to increased leak of lung cfDNA into blood. In addition, cfDNA methylation profiles do not directly reveal the underlying pathology; hence, findings reported here may not be specific to COVID-19 but rather reflect a universal response to respiratory viral infections beyond SARS-CoV-2.

# Materials and Methods

## Clinical samples

Samples of healthy controls and hospitalized patients were obtained from patients treated in these centers: Shaare Zedek

Medical Center, Jerusalem, Israel; and the Hebrew University-Hadassah Medical Center, Jerusalem, Israel. Hospitalized patients were recruited in a comprehensive manner in internal medicine and ICU departments, without exclusion criteria. Samples of asymptomatic/mild disease were obtained from hotel-quarantined who tested positive for COVID-19. Both asymptomatic/mild and hospitalized patient groups were recruited between May and October 2020. All donors serving as controls denied having any acute or major chronic illnesses or receiving any medications at the time of blood donation. Patient demographics, clinical data, and cfDNA data are detailed in Tables S6, S7, S8, and S9. Clinical parameters were extracted from patients directly or via their EMRs.

### Collection of blood samples and extraction of cfDNA

Blood samples were collected in either EDTA or STRECK tubes. For EDTA, tubes were centrifuged at 1,500$g$ for 10 min at 4°C within 4 h of collection. Plasma was removed and recentrifuged at 3,000$g$ for 10 min at 4°C to remove any remaining cells. STRECK tubes were processed within 10 d of collection and were processed in the same manner, at 24°C. Plasma was then stored at –80°C until assay. cfDNA was extracted using the QIAsymphony SP instrument and its dedicated QIAsymphony Circulating DNA Kit (QIAGEN) according to the manufacturer's instructions. DNA concentration was measured using Qubit dsDNA HS Assay Kit (Thermo Fisher Scientific). DNA derived from all samples was treated with bisulfite using EZ DNA Methylation-Gold (Zymo Research) and eluted in 24 $\mu$l elution buffer. gDNA was extracted directly from whole blood using QIAsymphony DNA Midi Kit (QIAGEN). Note that the gDNA content in such preparations is similar to the gDNA content of white blood cells, because the other components of whole blood—erythrocytes and platelets—contain negligible amounts of DNA.

### Whole-genome bisulfite sequencing and deconvolution

Dual-indexed sequencing libraries were prepared using Accel-NGS Methyl-Seq DNA library preparation kits (Swift BioSciences) and custom liquid handling scripts executed on the Hamilton MicroLab STAR. Libraries were quantified using KAPA Library Quantification Kits for Illumina Platforms (Kapa Biosystems). Four uniquely dual-indexed libraries, along with 10% PhiX v3 library (Illumina), were pooled and clustered on an Illumina NovaSeq 6000 S2 flow cell followed by 150-bp paired-end sequencing.

### Computational analysis of WGBS samples

Paired-end FASTQ files were mapped to the human genome (hg19) using bwa-meth (V 0.2.0) (Pedersen et al, 2014 Preprint). Duplicated reads were marked by Sambamba (V 0.6.5) (Tarasov et al, 2015). Reads with low (<10) mapping quality, duplicated, or not mapped in a proper pair were excluded. Reads were stripped from non-CpG nucleotides and converted to PAT files using wgbstools (V 0.1.0) (Loyfer et al, 2023).

We applied our previously published fragment-level deconvolution algorithm, UXM (Loyfer et al, 2023), to estimate

the cell-type composition of the COVID-19 (n = 6) and control (n = 6) plasma samples. The reference atlas includes 37 healthy cell types, and the features are 900 cell-type specifically unmethylated blocks. UXM software was used with default parameters.

### Assembly of targeted DNA methylation markers

Specific CpG sites were selected by examining WGBS data and identifying differentially methylated or differentially unmethylated regions, having at least four CpG sites within 150 bp. We selected regions showing methylation pattern with less than 0.3 in the specific cell type and greater than 0.8 in over 90% of other cells (Fig S1A) and designed primers to amplify ~120-bp fragments surrounding them using the multiplex two-step PCR amplification method (Neiman et al, 2020). Marker coordinates and primer sequences are provided in Table S10.

The validation of markers was done using DNA extracted from different cells and tissues, and the methylation status of the CpG block was assessed (Table S11). Some markers were more sensitive if one CpG site was allowed to be methylated differently than other CpGs in the block, as indicated in Table S10.

### PCR

To efficiently amplify and sequence multiple targets from bisulfite-treated cfDNA, we used a two-step multiplexed PCR protocol, as described previously (Neiman et al, 2020). Briefly, in the first step, up to 20 primer pairs were used in one PCR to amplify regions of interest from bisulfite-treated DNA, independent of the methylation status. Primers were 18–30 base pairs (bp) with primer melting temperature ranging from 58°C to 62°C. To maximize amplification efficiency and minimize primer interference, the primers were designed with additional 25-bp adaptors comprising Illumina TruSeq Universal Adaptors without index tags. All primers were mixed in two separate reaction tubes. For each sample, the PCR was prepared using the QIAGEN Multiplex PCR Kit according to the manufacturer's instructions with 7 $\mu$l of bisulfite-treated cfDNA. Reaction conditions for the first round of PCR were as follows: 95°C for 15 min, followed by 30 cycles of 95°C for 30 s, 57°C for 3 min, and 72°C for 1.5 min, followed by 10 min at 68°C.

In the second PCR step, the products of the first PCR were treated with exonuclease I (Thermo Fisher Scientific) for primer removal according to the manufacturer's instructions. Cleaned PCR products were amplified using one unique TruSeq Universal Adaptor primer pair per sample to add a unique index barcode to enable sample pooling for multiplex Illumina sequencing. The PCR was prepared using 2× PCRBIO HS Taq Mix Red Kit (PCR Biosystems) according to the manufacturer's instructions. Reaction conditions for the second round of PCR were as follows: 95°C for 2 min, followed by 15 cycles of 95°C for 30 s, 59°C for 1.5 min, 72°C for 30 s, followed by 10 min at 72°C. The PCR products were then pooled, run on 3% agarose gels with ethidium bromide staining, and extracted by the Zymo GEL Recovery kit.

### NGS analysis

As mentioned, cfDNA was PCR-amplified with primers specific for bisulfite-treated DNA but independent of the methylation status at the monitored CpG sites. Treatment with bisulfite led to degradation of 60–90% of the DNA (on average, 75% degradation), consistent with previous reports (Leontiou et al, 2015). Note that although DNA degradation does reduce assay sensitivity (because fewer DNA molecules are available for PCR amplification), it does not significantly harm assay specificity because methylated and unmethylated molecules are equally affected. Primers were barcoded using TruSeq Index Adapters (Illumina), allowing the mixing of samples from different individuals when sequencing PCR products using NextSeq sequencers (Illumina). Sequenced reads were separated by barcode, aligned to the target sequence, and analyzed using custom scripts written and implemented in R. Reads were quality-filtered based on Illumina quality scores and identified by having at least 80% similarity to target sequences and containing all the expected CpGs in the sequence. CpGs were considered methylated if CG was read and were considered unmethylated if TG was read. The fraction of unmethylated molecules in a sample was multiplied by the total concentration of cfDNA in the sample, to assess the number of specific cell genome equivalents per milliliter of plasma. The concentration of cfDNA was measured before bisulfite conversion, rendering the assay robust to potential intersample fluctuations in the extent of bisulfite-induced DNA degradation.

The computational pipeline used to interpret sequence reads was uploaded to GitHub (https://github.com/Joshmoss11/btseq; swh:1:rev:efc75ddd347c20392cf0a034706a7b5b6090be75).

### Plasma cfChIP-seq chromatin analysis

Plasma collection, cfChIP assay, and preprocessing of sequencing data were performed as previously described (Sadeh et al, 2021). Differential gene representation between hospitalized and asymptomatic samples was identified using a likelihood ratio test as described in the "Comparison of two groups of samples" section of the supplementary note (Sadeh et al, 2021). For the enrichment test of a predefined interferon response gene set, we discarded all genes with an average TSS coverage above 10 in an independent cohort of healthy samples that was not used in this study.

### Statistics

To assess the significance of differences between groups, we used a two-tailed Mann–Whitney $U$ test. We calculated all CIs at the 95% level, and $P$-value was considered significant when less than 0.05. In the correlation matrix, Pearson's and Spearman's tests were used to establish correlation between quantitative or qualitative parameters, respectively, and the FDR method was used to correct for multiple comparisons.

### Study approval

This study was conducted according to protocols approved by the institutional review boards at each study site: Hadassah-Hebrew University Medical Center, Jerusalem, Israel; Shaare Zedek Medical Center, Jerusalem, Israel; procedures were performed in accordance with the Declaration of Helsinki. Blood was obtained from donors who provided written informed consent. Case report forms were filled out by donors detailing underlying diseases.

### Code and data availability

WGBS analysis was performed using wgbstools (V 0.1.0) (https://github.com/nloyfer/wgbs_tools). Targeted bisulfite sequencing analysis was performed using btseq (https://github.com/Joshmoss11/btseq). Source data are provided with this article. The WGBS data generated in this study have been deposited in the GEO database under the accession code GSE302507. The targeted bisulfite sequencing data generated in this study are provided in the Supplementary Information.

## Supplementary Information

## Acknowledgements

We thank Idit Shiff and Abed Nasseredin from the Core Research Facility at the Hebrew University Faculty of Medicine for their support in sequencing analysis, and Noa Makhervax and Dr Lilach Gavish for help in coordinating the effort. This work was supported by a generous gift from Shlomo Kramer, supported by grants from Human Islet Research Network (HIRN UC4DK116274 and UC4DK104216 to R Shemer and Y Dor); Ernest and Bonnie Beutler Research Program of Excellence in Genomic Medicine, the Alex U Soyka pancreatic cancer fund, the Israel Science Foundation, the Waldholtz/Pakula family, the Robert M and Marilyn Sternberg Family Charitable Foundation, the Helmsley Charitable Trust, Grail and the DON Foundation (to Y Dor); and European Research Council grant (ERC Adg 101019560 to N Friedman). Y Dor holds the Walter and Greta Stiel Chair and Research grant in Heart studies. R Ben-Ami received a fellowship from the Rothschild Foundation. A Rokach received grant from the KAMLA Research Fund of the Hebrew University of Jerusalem and Shaare Zedek Medical Center.

### Author Contributions

R Ben-Ami: conceptualization, data curation, formal analysis, validation, investigation, visualization, methodology, and writing—original draft, review, and editing.
N Loyfer: data curation and visualization.
E Cohen: data curation.
G Fialkoff: formal analysis.
I Sharkia: formal analysis.
S Piyanzin: investigation.
N Bogot: resources.
D Kochan: resources.
G Kalak: resources.
A Jarjoui: resources.
C Chen-Shuali: resources.

H Azulai: resources.

H Barhoum: resources.

N Arish: resources.

MM Greenberger: resources.

D Velleman: resources.

R Kurd: resources.

E Ben-Chetrit: resources.

D Bohm: resources.

T Wolak: resources.

A Quteineh: resources.

G Cann: resources and software.

B Glaser: conceptualization and methodology.

N Friedman: conceptualization, supervision, investigation, and methodology.

T Kaplan: conceptualization, resources, software, and methodology.

R Shemer: conceptualization, supervision, methodology, and project administration.

A Rokach: conceptualization, resources, supervision, investigation, methodology, project administration, and writing—review and editing.

Y Dor: conceptualization, supervision, investigation, methodology, and writing—original draft, review, and editing.

## Conflict of Interest Statement

Supported in part by GRAIL, Inc., G Cann is an employee and shareholder at GRAIL, Inc. I Sharkia and N Friedman are shareholders and/or founders at Senseera, Inc. N Loyfer, T Kaplan, B Glaser, R Shemer, and Y Dor have filed patents on cfDNA analysis technology. The remaining authors have declared no conflict of interest.

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
