## [Reviewer comments · Life Science Alliance]

Life Science Alliance

Epigenetic liquid biopsies reveal endothelial turnover and erythropoiesis in asymptomatic COVID-19

Roni Ben-Ami, Netanel Loyfer, Eden Cohn, Gavriel Fialkoff, Israa Sharkia, Sheina Piyanzin, Naama Bogot, Danit Kochan, George Kalak, Amir Jarjoui, Chen Chen-Shuali, Hava Azulai, Hezi Barhoum, Nissim Arish, Moshe Greenberger, David Velleman, Ramzi Kurd, Eli Ben-Chetrit, Davina Bohm, Talya Wolak, Ahmad Quteineh, Gordon Cann, Benjamin Glaser, Nir Friedman, Tommy Kaplan, Ruth Shemer, Ariel Rokach, and Yuval Dor

DOI: <https://doi.org/10.26508/lsa.202503417>

Corresponding author(s): Yuval Dor, Hebrew University of Jerusalem and Danit Kochan, Shaare Zedek Medical Center

Review Timeline:

Submission Date:	2025-06-10
Editorial Decision:	2025-07-02
Revision Received:	2025-07-15
Accepted:	2025-07-16

Scientific Editor: Tim Fessenden

Transaction Report:

Please note that the manuscript was reviewed at *Review Commons* and these reports were taken into account in the decision-making process at *Life Science Alliance*.

Reviews

Review #1

The authors analyzed circulating cell-free DNA for COVID-19 using deep sequencing of the methylation and histone modification. The major output was cell-specific quantification. The study involved 120 unvaccinated, hospitalized patients, 19 asymptomatic/mild cases, and 40 controls. Between COVID-19 and controls, they found significant differences in lung epithelial cells, cardiomyocytes, vascular endothelial cells and erythroblasts. The latter two cell types had significant differences even in the asymptomatic patients. It is unclear if the damage seen is related to COVID-19 specifically, or related to general inflammation or infection.

Strengths of the study include relatively high WGBS/targeted sequencing, along with fragment-level analysis with methods described in their previous work (Loyfer et al. Nature). In addition, they add and ChIP-seq data using their published methods. The work comes from a group with leading expertise in methylation cell-free DNA analysis.

Overall, the work is most comprehensive analysis to date for COVID-19, and the data would be a valuable resource to the research community. We have major and minor comments that do not necessarily require additional experimental work.

Major comments:

1. There is a lack of data and the methods are presented in such a way that the results and conclusion can be reproduced and evaluated. Neither the code nor the data to generate the results are available. Both need to be made available during the peer review process.

Missing data: Fragment-level FASTQ, BAM, or PAT files are needed to reproduce the results.

Missing Scripts, for example in Github, is standard and reasonable for reproducing the figures shown.

Missing targeted assay method details:

- The authors should show the data, methods, and details for: "The validation of markers was done using DNA extracted from different cells and tissues, and the methylation status of the CpG block was assessed."

2. The authors did not list the major limitations of the study in the discussion or elsewhere. These should include (or be addressed with experimental or conclusion changes):

1. The small sample size of the asymptomatic/mild group (if the emphasis of the paper, as suggested by the title, is on the asymptomatic/mild group - see the next major point).

2. The targeted assay is used on the vast majority of samples, including all of the asymptomatic/mild group. However, it is limited to a particular subset of cell types (total defined by all possible cell types in the body). Those cell types were determined based on WGBS data on only 6 COVID-19 cases.

3. The methylation references for the WGBS data were limited to a fraction of all human cell types. For example, this paper was not able to evaluate Schwann cells or peripheral nerves, which was a significant finding for COVID-19 related multisystem inflammatory syndrome (PMID 37279751).

4. The case and control groups (severe, asymptomatic mild, and control) were collected at different times and circumstances, allowing for potential pre-analytical confounders.

5. cfDNA levels can be influenced by several unmeasured factors, including death, replication leading to more turnover, clearance/stability, and movement from tissue into circulation. The methods used cannot distinguish between these possibilities.

6. (if true) the controls used for the targeted assay were not age/sex matched. The median age for the controls skew younger per Table S1, S2, S3.

7. (optional) It is unclear whether the differences found are attributable to COVID-19, coronavirus infection, viral infections, infections in general, or inflammation in general. The appropriate alternative controls were not addressed in this study. The paper shows some degree of correlation with acute inflammatory markers (CRP, ferritin, neutrophil contribution).

3. The title is a bit misleading in that it revolves around the asymptomatic patients. However, this is also the group with the lowest representation at n=19. The vast majority of the data is related to the hospitalized patients. While other studies may have looked at hospitalized patients, I agree with the authors that there is merit in deep sequencing and

the correlated clinical data.

4. More details on the patient inclusion criteria are needed. Were the asymptomatic/mild positive by PCR test or a point of care immunoassay? We know the viral load is quite dynamic for these patients. What was the timing of the blood draw?

Likewise, how did you find the hospitalized patients? Was it comprehensive over a period of time? These details help reveal any potential biases in the selection process.

****Minor comments:****

1. The abstract states: "Asymptomatic patients had elevated levels of immune-derived cfDNA but did not show evidence of pulmonary or cardiac damage." However, in Fig 5, there seems to be a bimodal distribution for the lung epithelial and cardiomyocytes. Unclear if that is an artifact of the graph.

It is quite interesting that the asymptomatic/mild group seems to have a bimodal distribution in lung epithelial and cardiomyocyte cfDNA. Perhaps this data is not available, and the sample size is small, but could there have been a clinical difference between the two groups (e.g. asymptomatic versus mild, or had symptoms later?). It is unclear how precise the measurements are for the lung epithelial cells.

2. The authors listed 2 prior studies that looked at cell type or tissue damage during COVID-19. There are 2 other studies that I am aware of: PMID: 33651717 (n=84 with n=18 nonhospitalized) but probably shallow WGBS, and 37279751 (n=205 pediatric patients). Importantly, the latter paper found Schwann cells were significantly elevated, which is missing from the current study's assessment. In addition, citation 14 from the same group already found significantly increased vascular endothelial cfDNA in COVID-19 patients with severe disease versus mild. While some findings are consistent, there are also discrepancies.

3. Is Fig 2 necessary? Fig. 5 seems to display the same data but with the asymptomatic group.

4. "Elevated lung cfDNA reflects excessive lung cell death" - recommend this statement is tempered as direct evidence is not available in this study. An alternative explanation could be that endothelial cells are damaged, and it is easier for lung cfDNA to enter blood circulation rather than the respiratory system.

5. Fig 6: Add unit of measure to heatmap.

Supplemental Fig 1.: Add label to unit of measure in caption or figure. Average or median beta value over a series of CpGs?

6. The authors state the targeted assay "allows for a more accurate and sensitive detection of cfDNA from a given source", which should be tempered unless clear evidence is presented for these statements. In addition, it targets only a small subset of all cell types. The highest cell type contribution from MK cells is only represented by 2 markers.

7. Targeted assay has a few caveats that the authors should mention or fix: The method is not described in detail. Methods besides WGBS can have biases in methylation representation and a beta correlation between the 12 samples that underwent WGBS and the targeted assay would be reassuring. The level of precision at the lower end of cellular contribution would be helpful too. The lung epithelial and cardiomyocyte cells were present at the lower end of the spectrum. This can be shown in a titration of the purified cells into plasma, or at least an in silico titration analyzed with only the targeted markers.

8. The authors state "(i) Evidence of frequent cardiomyocyte death in hospitalized patients... it has not been appreciated that cardiac cell death is a feature shared by most hospitalized patients." However, COVID-19 patients have elevated troponin.

9. The authors state "This signal presumably reflects elevated turnover of red blood cells and increased rate of erythropoiesis". However, could it be also higher nucleated RBCs released into circulation as the authors cited?

10. Fig 2, 4, 5: The graphs seem to suggest that the authors picked 0.001 GE/mL as not detected. Should they label that point appropriately as "not detected" or "ND"? It is not clear why 0.001 GE/mL was picked, and the analytical sensitivity of the targeted test is not reported.

11. How many mLs of plasma were used?

General assessment:

Strengths

1. Interesting topic: Non-invasive tabulation using deep methylation sequencing of cell type shedding into circulation of an important disease (COVID-19).

2. Deep sequencing using methylation and histone output is a significant improvement on past studies. Although targeting limits the scope of the cell types, the targeting was based on relatively deep WGBS sequencing on 6 cases

and 6 controls.

Limitations

The unique aspects (targeted assay and deep sequencing) are missing data and detailed methodology for reanalysis and reproducibility.

See major comment 2.

Advance: The authors used deep sequencing through brute force (WGBS) and a unique targeted assay to study COVID-19 from a large group (n=120 patients). They found that endothelial and erythroblast lineages are overrepresented based on the presence and severity of the COVID-19 infection. Their findings are significant and go beyond what has been published. The methodologies and data (i.e. the controls) would be a great resource to the community that can be used beyond the scope of COVID-19.

Audience: This article would be appealing to a broad, translational/clinical audience. The authors have published on methylation deconvolution several times before, but to my knowledge, the broader targeted assay is unique and there is a large dataset with correlated clinical information that may be of broad utility.

Reviewer expertise: technical expertise with circulating cell-free DNA. translational/clinical expertise.

Review #2

In the manuscript entitled "Epigenetic liquid biopsies reveal elevated vascular endothelial cell turnover and erythropoiesis in asymptomatic COVID-19 patients," Ben-Ami and colleagues perform WGBS, targeted methylation assay and cfChIP-seq to measure cellular turnover/death or tissue injuries and infer gene expression profile in COVID-19 patients and healthy controls. First, they performed deep WGBS on severe COVID-19 and HC plasma samples, applied the novel UXM algorithm that includes 40 human cell types to identify the tissue origins of cfDNA, and showed increased cfDNA from diverse cell and tissue types in COVID-19 patients than healthy controls. Besides WGBS, they also performed targeted methylation assay to measure cellular turnovers/death and tissue injury from major cell and tissue types involved in COVID-19 pathogenesis and used as a predictor of poor outcome. Finally, they showed that cfChIP-seq can identify heightened immune responses associated with COVID-19 and asymptomatic patients. Previous studies have shown that cfDNA has a great potential to map tissue injuries in COVID-19 and predict patient outcomes (Cheng et al., 2021 & Andargie et al., 2021). The expanded reference methylation atlas and the addition of targeted methylation assay and cfCHIP-seq in this study are very informative and fascinating. Please allow me to congratulate Ben-Ami and colleagues for this wonderful work.

Below are some points that need to be addressed to improve the manuscript:

****Major****

1. Given the heterogeneous nature of COVID-19 clinical manifestation, the limited number of patients (n=6) raises concern about the significance of WGBS analysis. The authors need to provide further details as to why they performed WGBS only from 6 samples out of 120 subjects and what was the selection criteria.
2. The gene expression analysis with cfChIP-seq is interesting. Likewise, Differentially Methylated Regions (DMR) can infer gene expression. Is the methylation analysis also showing increased interferon response in COVID-19 patients? This study also showed increased cfDNA from monocytes that is not reflected in blood cell counts. Does cfChIP-seq identify inflammatory response-related genes in monocytes/macrophages? Hadjadj et al. 2020 (PMID: 32661059: Science) reported impaired interferon response in severe COVID-19 patients. Whereas this study showed heightened interferon response in severe and asymptomatic/mild COVID-19 patients compared to healthy controls, there was no difference between Mild and Severe COVID-19 patients. The author should consider validating their finding with plasma cytokine measurement. cfChIP-seq also identifies cfDNA tissues-of-origin (PMID: 33432199). How is the correlation between these three assays (WGBS, targeted methylation assay and cfChIP-seq) to detect cell death/turnover?
3. It is unclear whether hospitalized COVID-19 subjects experienced particular organ involvement. It would be interesting to link the tissue-specific cfDNA to different COVID-19 endotypes. For instance, cardiac involvement and cardiomyocyte cfDNA.
4. Previous studies reported cfDNA concentration in healthy controls ranges between 3 and 15 ng/mL. This study's median cfDNA level for asymptomatic COVID-19 patients falls within that range. It would be interesting if the authors

comment on the methodology differences, including plasma volume, correction for extraction efficiency, and cfDNA assay type.

5. Were the asymptomatic/Mild case samples collected in the same time frame as Hospitalized patients? It would be interesting if the authors comment on the effect of SARSCOV-2 variants and viral loads on plasma cfDNA level.
6. The author showed cfDNA from total T cells and CD8 cells in particular. The authors should comment on why CD4+ T was not shown instead of T cells (which includes both CD4 and CD8 cells).
7. Considering the expensive nature of deep sequencing, it would be interesting if the authors comment on applying the UXM algorithm for low and medium- and low-coverage sequenced samples.

****Minor****

1. The timing of blood sample collection from hospital admission or testing positive for COVID-19 is important to use cfDNA as a predictor of outcome. The authors should explain when the sample was collected for asymptomatic/mild patients. If it's not in the "acute phase," it should also be clarified for comparison with hospitalized COVID-19.
2. Is there a reason the authors included repeated measures of cfDNA within the same subject (N=120, n=142; Figure 1A)? The author should consider statistical correction for repeated measures. This is important to reduce bias.
3. I believe the authors forgot to include "Code and data availability" declaration. I encourage the authors to make publicly available the WGBS data and deconvolution algorithm for reproducibility purposes.
4. Figure 1D should show individual data points to see the pattern of tissue-specific cfDNA better, especially as COVID-19 shows heterogeneous clinical presentation. Please consider overlaying the data point on the histogram.
5. Methods - Page 27, the first sentences from the last paragraph, please include the unit.
6. after the number "75".... In fact, this paragraph is identical to the previous paper (PMID: 33432199); please consider paraphrasing the section.
7. Please clearly define "deteriorated." What WHO score or range is considered as deteriorated?
8. The authors mix between 40 and 37 reference cell types. Please be consistent.
9. Page 6, line 3, please replace erythrocyte with erythroblast.
10. Page 28, line 10, please replace COVID with COVID-19.
11. Figure 5D needs a key for recovered versus deteriorated.
12. Figure 5, legend title, please fix the number of healthy controls.... (n=30-45).

This manuscript used a deep WGBS approach with an expanded human cell-type methylation atlas and novel deconvolution algorithm, targeted methylation assay (which makes the cfDNA test easy to use in a clinical lab setting) and cfChIP-seq on plasma cfDNA based on epigenetic markers to identify specific cellular/organ involved in COVID-19 pathogenesis and identify potential mechanistic insights associated with heightened inflammatory response. Compared to the previous study, the limited sample size raises concerns about the significance of whole-genome bisulfite sequencing data in COVID-19 patients. Additionally, whether the tissue-specific cfDNA tracks specific COVID-19-associated endotypes has yet to be discussed. Taken together, this cfDNA may help to understand COVID-19 pathogenesis and define tissue or organ injuries.

My expertise is in Genomics and Immunology.

Detailed response to Reviewer comments

We thank the reviewers for their positive and constructive evaluation of the paper. We have addressed in full the concerns raised as detailed below. We apologize for the long time it took us to respond, which was a consequence of local circumstances in the last year.

Reviewer #1 (Evidence, reproducibility and clarity (Required)):

Summary:

The authors analyzed circulating cell-free DNA for COVID-19 using deep sequencing of the methylation and histone modification. The major output was cell-specific quantification. The study involved 120 unvaccinated, hospitalized patients, 19 asymptomatic/mild cases, and 40 controls. Between COVID-19 and controls, they found significant differences in lung epithelial cells, cardiomyocytes, vascular endothelial cells and erythroblasts. The latter two cell types had significant differences even in the asymptomatic patients. It is unclear if the damage seen is related to COVID-19 specifically, or related to general inflammation or infection.

Strengths of the study include relatively high WGBS/targeted sequencing, along with fragment-level analysis with methods described in their previous work (Loyfer et al. Nature). In addition, they add and ChIP-seq data using their published methods. The work comes from a group with leading expertise in methylation cell-free DNA analysis.

Overall, the work is most comprehensive analysis to date for COVID-19, and the data would be a valuable resource to the research community. We have major and minor comments that do not necessarily require additional experimental work.

We thank the reviewer for these supportive comments.

Major comments:

1. There is a lack of data and the methods are presented in such a way that the results and conclusion can be reproduced and evaluated. Neither the code nor the data to generate the results are available. Both need to be made available during the peer review process.

Missing data: Fragment-level FASTQ, BAM, or PAT files are needed to reproduce the results.

Missing Scripts, for example in Github, is standard and

reasonable for reproducing the figures shown.

Missing targeted assay method details:

-The authors should show the data, methods, and details for:
"The validation of markers was done using DNA extracted from different cells and tissues, and the methylation status of the CpG block was assessed".

Thank you. WGBS data files are currently being uploaded to GEO and are waiting for an accession number.

For the validation of targeted markers, we added a new supplemental table (S11) with data on the methylation status of the loci used in this study in different cells and tissues (i.e. marker specificity), and provided a detailed text and references to the methods used.

2. The authors did not list the major limitations of the study in the discussion or elsewhere. These should include (or be addressed with experimental or conclusion changes):

(1The small sample size of the asymptomatic/mild group (if the emphasis of the paper, as suggested by the title, is on the asymptomatic/mild group - see the next major point.

Thank you, indeed this is a limitation, we have now addressed this issue in the text. Despite this limitation, findings regarding to this population were statistically significant.

(2The targeted assay is used on the vast majority of samples, including all of the asymptomatic/mild group. However, it is limited to a particular subset of cell types (total defined by all possible cell types in the body). Those cell types were determined based on WGBS data on only 6 COVID-19 cases.

Thank you, indeed this is a limitation. WGBS was done on 6 critically ill patients, to uncover the potential cell types that will be of most interest in the targeted assay. In comparison to the WGBS, the targeted assay has a deeper coverage and therefore greater sensitivity. We have now addressed this issue in the limitations section.

(3The methylation references for the WGBS data were limited to a fraction of all human cell types. For example, this paper was not able to evaluate Schwann cells or peripheral nerves, which was a significant finding for COVID-19 related multisystem inflammatory syndrome (PMID 37279751).

The WGBS atlas (PMID: 36599988) consists of ~40 cell types that we were able to isolate at a high purity. While this is the most complete methylome atlas of human cell types generated to date, it is indeed incomplete. Unfortunately the scarcity of

Schwann cells prevented us from determining the methylome of this cell type, and the matter is to be investigated in future studies. Note that the study referred to by the reviewer described the cell-free transcriptome rather than the cfDNA methylome of patients. cfDNA methylation analysis of Schwann cells remains a challenge to be addressed in future studies. This limitation is explained in the revised text.

(4The case and control groups (severe, asymptomatic mild, and control) were collected at different times and circumstances, allowing for potential pre-analytical confounders.

We now addressed this limitation in the text.

(5cfDNA levels can be influenced by several unmeasured factors, including death, replication leading to more turnover, clearance/stability, and movement from tissue into circulation. The methods used cannot distinguish between these possibilities.

Indeed, the mechanism by which cfDNA concentration is increased is not fully understood, but is certainly correlated with pathology. We clarify this in the revised text.

) (6if true) the controls used for the targeted assay were not age/sex matched. The median age for the controls skew younger per Table S1, S2, S3.

We used control samples that were collected before the pandemic, to make sure that they were not infected with COVID-19. Consequently, there are minor demographic differences (e.g. controls tend to be younger than the hospitalized patients, though similar age to the asymptomatic donors).

Note that in previous studies, cfDNA levels and origins did not show differences in sex.

In the WGBS samples, we did age and sex matched the samples.

We explain this issue in the revised text.

) (7optional) It is unclear whether the differences found are attributable to COVID-19, coronavirus infection, viral infections, infections in general, or inflammation in general. The appropriate alternative controls were not addressed in this study. The paper shows some degree of correlation with acute inflammatory markers (CRP, ferritin, neutrophil contribution.(

Indeed, elevated cfDNA from specific tissues reflects tissue turnover or death, with no indication of the cause of pathology. We now addressed this limitation in the text.

.3The title is a bit misleading in that it revolves around the asymptomatic patients. However, this is also the group with the lowest representation at n=19. The vast majority of the data is related to the hospitalized patients. While other studies may have looked at hospitalized patients, I agree with the authors that there is merit in deep sequencing and the correlated clinical data.

Thank you. We chose to highlight in the title the most novel and provocative finding of the study.

.4More details on the patient inclusion criteria are needed. Were the asymptomatic/mild positive by PCR test or a point of care immunoassay? We know the viral load is quite dynamic for these patients. What was the timing of the blood draw?

Likewise, how did you find the hospitalized patients? Was it comprehensive over a period of time? These details help reveal any potential biases in the selection process.

We do not have information on the viral load in patients. All were positive for a PCR test. For the asymptomatic cases we know the time of the test, and this information is now added in Supplemental Table S2.

Hospitalized patients were recruited and consented at the Shaare Zedek Medical Center in a rather comprehensive manner – we recruited all patients that we could during May-June 2020. This is explained in the revised methods section.

Minor comments:

.1The abstract states: "Asymptomatic patients had elevated levels of immune-derived cfDNA but did not show evidence of pulmonary or cardiac damage." However, in Fig 5, there seems to be a bimodal distribution for the lung epithelial and cardiomyocytes. Unclear if that is an artifact of the graph.

It is quite interesting that the asymptomatic/mild group seems to have a bimodal distribution in lung epithelial and cardiomyocyte cfDNA. Perhaps this data is not available, and the sample size is

small, but could there have been a clinical difference between the two groups (e.g. asymptomatic versus mild, or had symptoms later?). It is unclear how precise the measurements are for the lung epithelial cells.

Thank you for this comment . Since cfDNA levels of the hospitalized patients are increased by orders of magnitude, we have arranged the graphs in logarithmic scale. Consequently, the bimodality that the reviewer mentions reflects only a slight absolute difference of cfDNA levels from lung and cardiomyocytes: ± 1 GE/ml, and we assume that this difference does not reflect clinical significance (and is not statistically different from the controls). This is referred to in the revised text.

.2The authors listed 2 prior studies that looked at cell type or tissue damage during COVID-19. There are 2 other studies that I am aware of: PMID: 33651717 (n=84 with n=18 nonhospitalized) but probably shallow WGBS, and 37279751 (n=205 pediatric patients). Importantly, the latter paper found Schwann cells were significantly elevated, which is missing from the current study's assessment. In addition, citation 14 from the same group already found significantly increased vascular endothelial cfDNA in COVID-19 patients with severe disease versus mild. While some findings are consistent, there are also discrepancies.

As explained above, our DNA methylation atlas does not contain a Schwann cell entry, so we cannot refer to cfDNA from this cell type; the mentioned study used cfRNA to assess this population. This is mentioned in the limitations of the study.

We now cite more comprehensively existing literature of liquid biopsies in Covid-19, and discuss the potential sources of discrepancy. We believe these result from differences in the methylome atlas, from the higher depth of the targeted assay compared with WGBS, and from our assessment of a unique population of asymptomatic patients.

Is Fig 2 necessary? Fig. 5 seems to display the same data but with the asymptomatic group.

Indeed there is some redundancy. Figure 2 shows data on hospitalized patients, and Figure 5 focuses on asymptomatic patients but uses as reference the same controls and severe patients as in Figure 2. We believe that this arrangement helps clarity.

" .4Elevated lung cfDNA reflects excessive lung cell death" - recommend this statement is tempered as direct evidence is not available in this study. An alternative explanation could be that endothelial cells are damaged, and it is easier for lung cfDNA to enter blood circulation rather than the respiratory system.

Thank you for this comment. We have addressed this possibility in the revised Discussion.

.5Fig 6: Add unit of measure to heatmap.

Added.

Supplemental Fig 1.: Add label to unit of measure in caption or figure. Average or median beta value over a series of CpGs?

Added. Each row represents a single CpG beta value.

.6The authors state the targeted assay "allows for a more accurate and sensitive detection of cfDNA from a given source", which should be tempered unless clear evidence is presented for these statements. In addition, it targets only a small subset of all cell types. The highest cell type contribution from MK cells is only represented by 2 markers

We now discuss this in more detail and with caution. Indeed targeted assays may not be more accurate given the use of few markers, but we do believe they are at least theoretically more sensitive given the use of PCR and deep sequencing.

.7Targeted assay has a few caveats that the authors should mention or fix:

The method is not described in detail.

More details are now provided, including multiplex PCR method and a reference to the script used for interpreting sequence data.

Methods besides WGBS can have biases in methylation representation and a beta correlation between the 12 samples that underwent WGBS and the targeted assay would be reassuring.

We have added a graph (new **Supplementary Figure S3**) showing a good correlation of Covid-19 WGBS data and targeted analysis of the same samples.

The level of precision at the lower end of cellular contribution would be helpful too. The lung epithelial and cardiomyocyte cells were present at the lower end of the spectrum. This can be shown in a titration of the purified cells into plasma, or at least an in silico titration analyzed with only the targeted markers.

Thank you. The targeted methylation assay is capable of detecting ~0.1% contribution of DNA from a given source, or 1-5 genome equivalents from this source. This is true also for our lung and cardiomyocyte markers, as previously shown (PMID 35450968, 29691397).

.8The authors state "(i) Evidence of frequent cardiomyocyte death in hospitalized patients... it has not been appreciated that cardiac cell death is a feature shared by most hospitalized patients." However, COVID-19 patients have elevated troponin.

Thank you. Evidence for troponin elevation was indeed reported in some, but not most of the hospitalized patients (see PMID: 32652195, 33121710, 32219356, 32211816). Note that troponin is not a definitive evidence of cardiac cell death (e.g. the significance of elevated troponin after a marathon or in patients with kidney disease is not clear). This provides a justification for the use of cfDNA for this purpose, as we have shown previously (PMID: 37290439). This is clarified in the revised Discussion.

.9The authors state "This signal presumably reflects elevated turnover of red blood cells and increased rate of erythropoiesis". However, could it be also higher nucleated RBCs released into circulation as the authors cited?

Thank you. Both of these possibilities are valid, and are not mutually exclusive. Elevated NRBC was reported in severe COVID-19, and is strongly associated with higher erythropoiesis. This is clarified in the revised Discussion.

.10Fig 2, 4, 5: The graphs seem to suggest that the authors picked 0.001 GE/mL as not detected. Should they label that point

appropriately as "not detected" or "ND"? It is not clear why 0.001 GE/mL was picked, and the analytical sensitivity of the targeted test is not reported.

Right, this was due to the non-zero limit of log graphs. We explain this in the text.

.11How many mLs of plasma were used?

We have now added to supplemental tables the amount of plasma that was used for each patient.

Reviewer #1 (Significance (Required)):

-General assessment:

Strengths-

(1Interesting topic: Non-invasive tabulation using deep methylation sequencing of cell type shedding into circulation of an important disease (COVID-19.)

(2Deep sequencing using methylation and histone output is a significant improvement on past studies. Although targeting limits the scope of the cell types, the targeting was based on relatively deep WGBS sequencing on 6 cases and 6 controls.

Limitations-

The unique aspects (targeted assay and deep sequencing) are missing data and detailed methodology for reanalysis and reproducibility.

See major comment 2.

-Advance: The authors used deep sequencing through brute force (WGBS) and a unique targeted assay to study COVID-19 from a large group (n=120 patients). They found that endothelial and erythroblast lineages are overrepresented based on the presence and severity of the COVID-19 infection. Their findings are significant and go beyond what has been published. The methodologies and data (i.e. the controls) would be a great resource to the community that can be used beyond the scope of COVID-19.

-Audience: This article would be appealing to a broad, translational/clinical audience. The authors have published on methylation deconvolution several times before, but to my knowledge, the broader targeted assay is unique and there is a large dataset with correlated clinical information that may be of broad utility.

-Reviewer expertise: technical expertise with circulating cell-free DNA. translational/clinical expertise.

Reviewer #2 (Evidence, reproducibility and clarity (Required)):

they performed deep WGBS on severe COVID-19 and HC plasma samples, applied the novel UXM algorithm that includes 40 human cell types to identify the tissue origins of cfDNA, and showed increased cfDNA from diverse cell and tissue types in COVID-19 patients than healthy controls. Besides WGBS, they also performed targeted methylation assay to measure cellular turnovers/death and tissue injury from major cell and tissue types involved in COVID-19 pathogenesis and used as a predictor of poor outcome. Finally, they showed that cfChIP-seq can identify heightened immune responses associated with COVID-19 and asymptomatic patients. Previous studies have shown that cfDNA has a great potential to map tissue injuries in COVID-19 and predict patient outcomes (Cheng et al., 2021 & Andargie et al., 2021). The expanded reference methylation atlas and the addition of targeted methylation assay and cfCHIP-seq in this study are very informative and fascinating. Please allow me to congratulate Ben-Ami and colleagues for this wonderful work.

Thank you for this encouraging feedback.

Below are some points that need to be addressed to improve the manuscript:

Major

.1 Given the heterogeneous nature of COVID-19 clinical manifestation, the limited number of patients (n=6) raises concern about the significance of WGBS analysis. The authors need to provide further details as to why they performed WGBS only from 6 samples out of 120 subjects and what was the selection criteria

Study design was impacted by resource limitations. We were able to perform deep WGBS only on a small number of samples, so have used this as a guide to the general nature of tissue turnover in COVID-19 patients, and later used a narrower, highly sensitive, more affordable and more broadly available targeted assay. This is clarified in the revised text (Discussion, section on limitations of study).

.2 The gene expression analysis with cfChIP-seq is interesting. Likewise, Differentially Methylated Regions (DMR) can infer gene

expression. Is the methylation analysis also showing increased interferon response in COVID-19 patients? This study also showed increased cfDNA from monocytes that is not reflected in blood cell counts. Does cfCHIP-seq identify inflammatory response-related genes in monocytes/macrophages? Hadjadj et al. 2020 (PMID: 32661059: Science) reported impaired interferon response in severe COVID-19 patients. Whereas this study showed heightened interferon response in severe and asymptomatic/mild COVID-19 patients compared to healthy controls, there was no difference between Mild and Severe COVID-19 patients. The author should consider validating their finding with plasma cytokine measurement. cfChip-seq also identifies cfDNA tissues-of-origin (PMID: 33432199). How is the correlation between these three assays (WGBS, targeted methylation assay and cfCHIP-seq) to detect cell death/turnover?

- Thank you for these comments. While cfChip does indeed reflect gene expression patterns in the cells that released cfDNA, cfDNA methylation patterns are indicative of cell identity (i.e. tissue of origin) but not dynamic gene expression (PMID: 30100054).

Unfortunately, current cfChip technology while revealing gene expression patterns in the cells that released cfChromatin, does not inform which cell types have expressed these genes (e.g. monocytes or T cells). Thus we can state that the cells releasing cfDNA expressed interferon stimulated genes, but we cannot say which cells were expressing these genes.

We were unable to perform additional measurements e.g. cytokines since our blood samples are almost entirely depleted.

With regards to the tissue origins of cfDNA: as shown in the paper, there is a general good agreement between WGBS and the targeted assay. In the revised version we show a good correlation between findings in specific samples that were subject to both WGBS and the targeted assay (Supplemental Figure S3). In our hands the sensitivity and specificity of cfChip-seq for detecting tissue origins of cfDNA are lower than cfDNA methylation, hence we elected to use the cfChip information only for inference of gene expression.

.3It is unclear whether hospitalized COVID-19 subjects experienced particular organ involvement. It would be interesting to link the tissue-specific cfDNA to different COVID-19 endotypes. For instance, cardiac involvement and cardiomyocyte cfDNA.

Indeed, linking tissue-specific cfDNA to clinical phenotype has been challenging. Elevated lung cfDNA is correlated with disease severity (which is well established to be associated with pulmonary damage). We were unable to link elevated cardiac cfDNA to a clinical cardiac

phenotype, also because of the limited cardiac assays that were performed on the hospitalized patients e.g troponin and cardiac eco.

.4 Previous studies reported cfDNA concentration in healthy controls ranges between 3 and 15 ng/mL. This study's median cfDNA level for asymptomatic COVID-19 patients falls within that range. It would be interesting if the authors comment on the methodology differences, including plasma volume, correction for extraction efficiency, and cfDNA assay type.

Indeed, asymptomatic patients had a mild, though highly statistically significant elevation in total cfDNA concentration relative to controls, as shown in Figure 5. Samples of asymptomatic patients and controls were obtained and processed identically using the Qiasymphony liquid handling robot. This is described in the revised methods. Plasma volume collected for each sample is now shown in Supp Tables S1-4.

.5 Were the asymptomatic/Mild case samples collected in the same time frame as Hospitalized patients? It would be interesting if the authors comment on the effect of SARSCOV-2 variants and viral loads on plasma cfDNA level.

Yes, all collected at the same period (May-October 2020). This is stated in the revised methods. Unfortunately we do not have information on specific variants on viral loads.

.6 The author showed cfDNA from total T cells and CD8 cells in particular. The authors should comment on why CD4+ T was not shown instead of T cells (which includes both CD4 and CD8 cells).

Unfortunately our current methylome atlas does not allow for identification of specific methylation markers for CD4+ cells (PMID: 34842142).

.7 Considering the expensive nature of deep sequencing, it would be interesting if the authors comment on applying the UXM algorithm for low and medium- and low-coverage sequenced samples.

The algorithm applies to WGBS samples regardless of depth, obviously with reduced performance in low coverage sequencing. Formal analysis of performance on multiple WGBS samples is ongoing.

Minor

.1The timing of blood sample collection from hospital admission or testing positive for COVID-19 is important to use cfDNA as a predictor of outcome. The authors should explain when the sample was collected for asymptomatic/mild patients. If it's not in the "acute phase" it should also be clarified for comparison with hospitalized COVID-19.

We have now added the time of sampling – typically a week or two after diagnosis (Supplemental Table S2).

.2Is there a reason the authors included repeated measures of cfDNA within the same subject (N=120, n=142; Figure 1A)? The author should consider statistical correction for repeated measures. This is important to reduce bias.

Thank you, we have now reanalyzed the data including only one sample for each patient. The results are largely the same as the original analysis (for reviewer eyes only).

.3I believe the authors forgot to include "Code and data availability" declaration. I encourage the authors to make publicly available the WGBS data and deconvolution algorithm for reproducibility purposes.

WGBS data files are currently being uploaded to GEO and are waiting for an accession number.

.4Figure 1D should show individual data points to see the pattern of tissue-specific cfDNA better, especially as COVID-19 shows heterogeneous clinical presentation. Please consider overlaying the data point on the histogram.

Thank you, we have changed the graph to show each datapoint.

.5Methods - Page 27, the first sentences from the last paragraph, please include the unit

Thank you, we have changed the paragraph.

.6after the number "75".... In fact, this paragraph is identical to the previous paper (PMID: 33432199); please consider paraphrasing the section.

Done.

.7Please clearly define "deteriorated." What WHO score or range is considered as deteriorated?

Deteriorated patients were defined as [maximal WHO score post sample] – [WHO score at sampling day] > 0. This is now clarified in the revised results section.

.8The authors mix between 40 and 37 reference cell types. Please be consistent.

Thank you. Done.

.9Page 6, line 3, please replace erythrocyte with erythroblast.

Done.

.10Page 28, line 10, please replace COVID with COVID-19.

Done.

.11Figure 5D needs a key for recovered versus deteriorated.

Done (figure 4D).

.12Figure 5, legend title, please fix the number of healthy controls.... (n=30-45).

Done.

Reviewer #2 (Significance (Required)):

This manuscript used a deep WGBS approach with an expanded human cell-type methylation atlas and novel deconvolution algorithm, targeted methylation assay (which makes the cfDNA test easy to use in a clinical lab setting) and cfChIP-seq on plasma cfDNA based on epigenetic markers to identify specific cellular/organ involved in COVID-19 pathogenesis and identify potential mechanistic insights associated with heightened inflammatory response. Compared to the previous study, the limited sample size raises concerns about the significance of whole-genome bisulfite sequencing data in COVID-19 patients. Additionally, whether the tissue-specific cfDNA tracks specific COVID-19-associated endotypes has yet to be discussed. Taken together, this cfDNA may help to understand COVID-19 pathogenesis and define tissue or organ injuries.

My expertise is in Genomics and Immunology.

July 2, 2025

RE: Life Science Alliance Manuscript #LSA-2025-03417

Prof. Yuval Dor
Hebrew University of Jerusalem
Cellular Biochemistry and Human Genetics
Ein Kerem Campus
Jerusalem, MA 91120
ISRAEL

Dear Dr. Dor,

Thank you for submitting your revised manuscript entitled "Epigenetic liquid biopsies reveal elevated vascular endothelial cell turnover and erythropoiesis in asymptomatic COVID-19 patients". While Reviewer 2 was unavailable, Reviewer 1 is satisfied that their concerns were addressed. We would be happy to publish your paper in Life Science Alliance pending additions to the text to address the two remaining minor concerns of this reviewer, and final revisions necessary to meet our formatting guidelines.

- Please be sure that the authorship listing and order is correct.
- Please upload your main manuscript text as an editable doc file.
- Please upload your main and supplementary figures as single files.
- Please add a Running Title and a Summary Blurb/Alternate Abstract in our system
- Please add a Category for your manuscript in our system.
- Please add the X and Bluesky handles of your host institute/organization, as well as your own and/or one of the authors, in our system.
- Please be sure that the authorship listing and order are correct and match between the system and the manuscript file.
- Please add the authors' contribution section to our system as well.
- Please consult our manuscript preparation guidelines <https://www.life-science-alliance.org/manuscript-prep> and make sure your manuscript sections are in the correct order.
- Please add your main, supplementary figure, and table legends to the main manuscript text after the references section.
- Please label the panels in Figure S1 to match their legend.
- Please remove the label of panel A from Figure S2 since this figure has only one panel.
- Please add callouts for Figures 4D; 6B and S1A-B to your main manuscript text
- Your manuscript currently contains both a Conflict of Interest statement and a Declaration of Interest statement, which are inconsistent with each other. Please revise to include only a Conflict of Interests statement and follow journal guidelines for its placement.
- The current Data Availability statement is incomplete. We discourage making data available "upon request" rather than having it publicly available, but if there is no alternative for privacy or other reasons, please describe what the data are and why they are not public, whom to contact (with a public email address), and conditions for re-use.

A. FINAL FILES:

B. MANUSCRIPT ORGANIZATION AND FORMATTING:

Sincerely,

Reviewer #1 (Comments to the Authors (Required)):

The authors addressed my previous comments. I have the following question for the authors to comment on.

1. It is well-documented that hospitalized COVID-19 patients often experience lymphocytopenia. This study showed increased cfDNA from lymphocytes, including B cells, T cells, and NK cells, similar to myeloid cells (neutrophils, monocytes/macrophages, megakaryocytes). In figure 3, unlike myeloid cells and their respective circulating cell counts, lymphocyte cfDNA negatively correlated with absolute or % circulating B cells, T cells and CD8 cells. Could the authors comment why?

2. How is the increased erythroblast and endothelial cfDNA in asymptomatic patients relevant for clinical practice? Is the endothelial cfDNA correlated with d-dimer in asymptomatic patients? Have you examined nucleated RBCs and hemoglobin levels in asymptomatic patients?

July 16, 2025

RE: Life Science Alliance Manuscript #LSA-2025-03417R

Prof. Yuval Dor
Hebrew University of Jerusalem
Cellular Biochemistry and Human Genetics
Ein Kerem Campus
Jerusalem, MA 91120
Israel

Dear Dr. Dor,

Thank you for submitting your Research Article entitled "Epigenetic liquid biopsies reveal endothelial turnover and erythropoiesis in asymptomatic COVID-19". It is a pleasure to let you know that your manuscript is now accepted for publication in Life Science Alliance. Congratulations on this interesting work.

DISTRIBUTION OF MATERIALS:

Again, congratulations on a very nice paper. I hope you found the review process to be constructive and are pleased with how the manuscript was handled editorially. We look forward to future exciting submissions from your lab.

Sincerely,
